# Inoculation and colonization of the entomopathogenic fungi, *Isaria javanica* and *Purpureocillium lilacinum*, in tomato plants, and their effect on seedling growth, mortality and adult emergence of Bemisia *tabaci* (Gennadius)

Ibrahim Sani[1,2], Syari Jamian[1,3]*, Norsazilawati Saad[1], Sumaiyah Abdullah[1], Erneeza Mohd Hata[4], Johari Jalinas[5], Siti Izera Ismail[1,3]*

1 Department of Plant Protection, Faculty of Agriculture, Universiti Putra Malaysia, Serdang, Malaysia, 2 Department of Biology, Faculty of Natural and Applied Sciences, Umaru Musa Yar'adua University, P.M.B., Katsina, Nigeria, 3 Laboratory of Climate-Smart Food Crop Production, Institute of Tropical Agriculture and Food Security (ITAFoS), Universiti Putra Malaysia, Serdang, Malaysia, 4 Laboratory of Sustainable Agronomy & Crop Protection, Institute of Plantation Studies, Universiti Putra Malaysia, Serdang, Selangor, 5 Centre for Insect Systematics, Department of Biological Sciences and Biotechnology, Faculty of Science & Technology, Universiti Kebangsaan Malaysia, Bangi, Malaysia

* syari@upm.edu.my (SJ); izera@upm.edu.my (SII)

## Abstract

Entomopathogenic fungi (EPF) are natural enemies which affect insect population and have long been recognized as biological control agents against many insect pests. Some isolates have also been established as endophytes, benefiting their host plants without causing any symptoms or negative effects. Here we demonstrated two entomopathogenic fungal species, *Isariajavanica* (Frieder. & Bally) Samson & Hywel-jone 2005 and *Purpureocillium lilacinum* (Thom) Luangsa-ard, Hou-braken, Hywel-Jones & Samson (2011) as endophytes in tomato plants by using the seed inoculation method and examined their effect on plant growth, *B. tabaci* mortality, and adult emergence. Our study indicated that tomato seeds treated with a fungal suspension of *I. javanica* and *P. lilacinum* enabled their recovery from plant tissues (root, stem and leaf) up to 60 days after inoculation (DAI). Both endophytic isolates also caused significant mortality of adult *B. tabaci* on seedlings inoculated with, *I. javanica* (51.92±4.78%), and *P. lilacinum* (45.32±0.20%) compared to the control treatment (19.29±2.35). Adult emergence rates were significantly high in the control treatments (57.50 ±2.66%) compared to *I. javanica* (15.00±1.47%) and *P. lilacinum* (28.75±4.78%) treatments. This study provides evidence that endophytic isolates of *I. javanica* and *P. lilacinum* have a biocontrol potentials for used against whiteflies and could also explored as plant growth promoters.

**Data Availability Statement:** All relevant data are within the paper and its Supporting information files.

**Funding:** This study was funded by the Fundamental Research Grant Scheme (FRGS) from the Ministry of Higher Education Malaysia (Grant no. 5540212; reference code FRGS/1/2019/WAB01/UPM/02/36). The funders had no role in study design, data collection and analysis, decision to publish, or preparation of the manuscript.

**Competing interests:** The authors have declared that no competing interests exist.

## Introduction

Entomopathogenic fungi (EPF) are natural enemies of the insect populations and are often regarded as effective biological control agents in integrated pest management (IPM) programs [1, 2]. Insect pests are normally controlled with EPF by either inundative or inoculative application of fungal propagules [3]. Several EPF have been commercialized for sap-sucking insect pest control [4]. In addition to direct applications, EPF have been discovered to colonize plants and grow endophytically, causing significant mortality of insect pests infesting plants, including *B. tabaci* on tomato plants [5].

In many different plants, numerous genera of EPF have been isolated as endophytes. Some of them were identified as endophytes that occur naturally while others have been introduced into the plants artificially using different methods of inoculation [6]. Ascomycete fungi such as *Beauveria bassiana* and *Metarhizium anisopliae* are typical examples of fungal endophytes that naturally colonize plants and have also been artificially introduced into a wide variety of plants to reduce damage caused by insect pests including leaf-mining insects, and sap-sucking pests [4, 7–10]. For example, the endophytic colonization of tomato by *B. bassiana* can significantly suppress *Tuta obsoluta* populations causing approximately 90% mortality [7, 11].

In recent years, EPF species from the genera *Isaria* and *Purpureocillium* have been reported to artificially colonized plant tissues, promoting plant growth and mortality of insect pests [12, 13]. Fungal-plant interactions have been demonstrated to last for several months after inoculation, protecting the plants against insect attacks [11, 14]. Depending on the inoculation techniques, endophytes vary in their ability to colonize various plant tissues and survive over a plant growth cycle [10]. Inoculation techniques include seed treatment, root dipping, stem injection soil drenching, and foliar application [15]. A range of these techniques has been studied for the endophytic colonization of tomato plants and efficiently promoting seedlings' growth and protecting them from attack by several insect pests.

In Malaysia where this study was conducted, little or no information can be found on the effects of EPF endophytes on the host plants and insect herbivores. However, there are some reports of endophytic colonization of some plants by other fungal isolates [16, 17], but no reports are yet available on the EPF establishment as an endophyte in tomatoes and any other plants.

The use of EPF as endophytes in plants could reduce damage caused by insect pests especially the whitefly, *Bemisia tabaci*, which has recently become one of the major threats to tomato production in Malaysia [18, 19]. Therefore, the present study was conducted to evaluate the endophytic capability in tomato plants by some recently isolated native strains of EPF in Malaysia and their influence on plant growth, mortality, and adult emergence of *B. tabaci* that could become part of IPM.

## Materials and methods

### Plants, insects, and endophytic fungal isolates

Tomato seeds (MT1 variety) used in this study were obtained from the Malaysian Agricultural Research and Development Institute (MARDI). The stock culture of whiteflies that colonized tomato plants in the greenhouse was collected, reared, and maintained in a greenhouse under 25–30°C and 60–80% RH with a photoperiod of 12L:12D h light: dark, at the Faculty of Agriculture, Universiti Putra Malaysia (UPM). *B. tabaci* was identified by using mitochondrial COI (mt COI) sequencing with the primers (C1-J-2195 and L2-N-3014) as described by Frohlich et al. [20]. The sequencing data was registered and assigned the accession number OM638559 (GenBank). *B. tabaci* populations were reared on tomato plants for over 10

generations in a rearing cage at the Ladang 15, Faculty of Agriculture, Universiti Putra Malaysia(UPM) at (25–30°C and 60–80% RH with a photoperiod of 12L:12D h. Tomato plants were checked daily, and the damaged seedlings were replaced with new seedlings. The two fungal isolates, *I. javanica* (Cjc-03) and *P. lilacinum* (TS-01) used in this study were found naturally infesting *B. tabaci* on chili and tomato plants and their virulence was evaluated against nymphs and adult *B. tabaci* under laboratory and glasshouse conditions. The isolates were identified by morphological characterization and molecular identification (ITS region amplification).

The sequence data of the 2 isolates have been deposited in the NCBI GenBank database and the details of the isolates can be seen in S1 Table.

## Preparation of conidial suspensions

The isolates were sub-cultured on potato dextrose agar medium (PDA) medium supplemented with 50mg/ml each of penicillin, streptomycin, and tetracycline. Conidia were harvested from 14 days old cultures grown on PDA slants by flooding 0.01% of tween 80 on sporulated cultures and scrapped using a sterile spatula. Following that, the collected conidia were suspended in a 50 mL plastic tube, vortexed for 2 minutes, and filtered through four layers of muslin cloth to remove debris and create a homogeneous stock suspension. Conidia viability was determined by inoculating 150µl of each fungal isolate suspension on to the surface of fresh PDA plates [21]. Before incubation, a sterile microscope coverslip was placed on top of the agar on each plate. At 18–20 hours after incubation, the viability assessment was conducted using a light microscope at 400X magnifications by counting the number of germinated conidia out of 100 conidia in one randomly selected area. When the diameter of the conidium's germ tubes exceeded half of the diameter of the conidium, the conidia were considered viable [10]. For each isolate, an average of three replicate counts was used.

## Tomato seed inoculation

Aseed immersion method was performed in this experiment by following the method described by Allegrucci et al. [7]. Seeds were surface sterilized using 70% ethanol for 5 minutes and rinsed 5 times with sterile distilled water. The treated seeds were placed on sterile tissue filter papers to dry for 30 minutes in a laminar flow before being inoculated and planted [7, 11, 22]. About 10 g of surface-sterilized seeds were subdivided into three equal portions, out of which two portions were individually soaked in a tube containing 10 ml of a conidial suspension with $1 \times 10^8$ conidia/ ml prepared from each of the two isolates (Cjc-03 and TS-01). The third portion was soaked in sterile distilled water +0.01% Tween 80 solution that served as a control treatment. At 24 hours after inoculation, the seeds were dried on sterile tissue paper in a laminar flow cabinet for 30 min, sown in multipots trays at a depth of 2 cm, containing approximately 20g of sterile peat moss planting media (previously autoclaved at 121°C for 30 minutes). The trays were maintained in the greenhouse at 25–30° C and 60–80%RH. During the experiments, plants were watered as needed and no fertilizer was used.

## Assessment of endophytic colonization in inoculated tomato seedlings

Endophytic colonization was examined using two methods: (i) surface sterilization and PDA media plating, and (ii) live imaging using confocal laser-scanning microscopy (CLSM). For the first method, at each observation point, 5 seedlings were randomly selected per treatment for endophytic colonization assessment using a destructive sampling method [23]. The seedlings were then transported to the laboratory, washed, separated into leaves, stems, and roots, and trimmed into smaller pieces of about 1 cm$^2$ with sterile stainless scissors. These parts were surface sterilized with 70% ethanol for 5 min, rinsed 5 times in sterile distilled water, and then

placed on sterile tissue paper to dry in a laminar flow cabinet [22]. Five pieces of each leaf, stem, and root from each inoculated seedling were randomly taken and placed in a PDA plate, and each of the plates was labeled and replicated 3 times.

The Petri dishes containing the plant tissues were then incubated at 25±2 ˚C and inspected daily to assess for fungal growth. Plant pieces that exhibited fungal growth were recorded. A total of 60 plants and 180 plant pieces were examined and the presence or absence of fungal endophytes in leaves, stems, and roots was recorded after 15, 30, 45, and 60 DAI.

Re-isolated fungal colonies were morphologically identified by comparing the mycelia and growth pattern with the original culture and by observing the conidia and conidiophores from prepared slides using a light microscope.

The data were expressed as:

$$\text{Colonization frequency} = \left( \frac{\text{Number of Plant Pieces Colonized}}{\text{Total Number of Plant pieces}} \right) \text{x } 100$$

The second method, CLSM live imaging, was carried out as previously described by Nishi et al. [9]. A sterile stainless steel scissor was used to trim the sample of plant tissue to fit a glass microscope slide. CLSM was used to observed samples within 1–2 hours of collection. About fifteen different plant parts were observed from each treatment. Selected images wererepresentative of all images taken for a particular sample type.

## Assessment of the effects of endophytic entomopathogenic fungi on tomato plant growth

The plant growth parameters: plant height, fresh weight, and dry weight were evaluated from the tomato plants used for endophytic colonization as described in the above section. The height of the tomato plants were measured with a ruler at 15, 30, 45, and 60 DAI. At 60 DAI, five seedlings from each of the treated and untreated control plants were harvested and taken to the laboratory for assessment of fresh and dry weight of the roots and aerial parts (stem and leaves). The fresh weight of the aerial parts and roots were weighed with an electronic balance and the same plant parts were placed inside a 20x20cm paper bag and placed in the oven at 60˚C. After 3 days, the dried aerial and root parts were measured using the same electronic balance [24].

## Effect of endophytic entomopathogenic fungi on *B. tabaci* mortality and adult emergence

The hypothesis tested in this experiment was that tomato seedlings inoculated as seeds with two EPF isolates, Cjc-03 and TS-01 have the ability to kill *B. tabaci* fed on colonized tomato plants. At 21 DAI, 6 seedlings from each treatment were transplanted in a 250ml pot (S1 Fig). A transparent plastic container (15 cm height X 8 cm width) with fine mesh at the top was placed inside the rim of pots covering the aerial part of the plant (S1 Fig). At 72 hours after transplanting, adult whiteflies were collected from a rearing cage with a manual aspirator and exposed to the aerial part of seedlings that had been grown from seeds inoculated with conidia of either *I. javanica* or *P. lilacinum* and untreated control seedlings. Afterward, 20–25 adult *B. tabaci* ($\leq$ 5 days) were released in each confined pot with seedlings and adults were allowed to feed and oviposit for 7 days. The mortality of *B. tabaci* were recorded by counting the number of dead adults that dropped off the plantin each cage. At 7 days post *B. tabaci* release, all surviving adult insects were removed from the aerial parts of treated and control plants. Eggs laid on the leaflets were observed and seedlings were transferred to a new cage (20 cm height x10 cm width) for observing emergence of the first-generation adults (S1 Fig). The eggs and

subsequent stages were monitored over a period of 35 days, while quantifying the number of first-generation adult *B. tabaci*.

## Data analysis

The data of plant height were analyzed using factorial analysis of the variance, while the data for endophytic colonization, fresh and dry weight, population, and adult emergence of *B. tabaci* were analyzed with a one-way analysis of variance (ANOVA) using the package *R* statistical software Version 3.6.1, and significant differences between the means were compared with the LSD-Fischer test ($p = 0.05$).

## Results

### Assessment of endophytic colonization of inoculated tomato seedlings

The result of this study on endophytic colonization of tomatoes showed that both of the EPF isolates examined (Cjc-03 and TS-01) successfully developed as endophytes in these plants (re-isolated from leaves, stem, and root) (S2 Fig) following seed treatment. There was a decrease in colonization as assessed from the plant tissues plated on PDA medium after 30 days of inoculation, however, no *P. lilacinum* was re-isolated at 60 days after inoculation (Fig 1). No fungal isolates were isolated from the control plants.

The colonization frequency observed for both two isolates showed higher colonization of the stems and leaves at 15 and 30 days of assessment respectively and showed no significant difference of colonization for each assessment period (15, 30 and 60 DAI), but significantly differed in the day interval (stem: F = 0.31 $p = 0.60$; F = 5.15 $p = 0.029$: leaves: F = 4.35, $p = 0.06$; F = 17.58, $p = 0.0005$) (Fig 1A and 1B). On the other hand, the highest root colonization by two fungal isolates was recorded at 30 days and significantly differed at each day of assessment (15, 30 and 60 DAI) (F = 5.21 $p = 0.045$; F = 14.15, $p = 0.001$) (Fig 1C). Endophytic colonization of both isolates (Cjc-03 and TS-01) in tomato plants were also examined using live imaging by CLSM. Spores, germinating conidia and appressorium formation were observed indicating that the conidia germinated in the tomato plant tissues (S3 Fig).

### Effect of Endophytic entomopathogenic fungi on tomato seedling growth

The effects of inoculated EPF on the growth of tomato plants; height, number of leaves (leaflets), and fresh and dry weight are presented in Figs 2 and 3. Data analyzed showed a significant increase of these parameters for tomato plants inoculated with EPF as compared to the control plant. At 15, 30, and 60 DAI, the following mean plant heights ± SE were recorded respectively: Cjc-03; 3.56 cm, 13.76 cm, 45.04 cm: TS-01; 5.90 cm, 14.06 cm, 46.28 cm (Fig 2A). The factorial analysis of variance showed significant effects of the fungal endophytes, day after inoculation and their interaction on the height of tomato seedling (F =. 32.93, df = 2,32, $p < 0.001$; F = 820.91, df = 2,32, $p < 0.001$; F = 9.64, df = 4,32, $p < 0.01$ (Fig 2A).

The leaves (leaflets) number of tomato seedlings showed a significantly higher mean number when seeds were inoculated with Cjc-03 and TS-01 isolates (74.67and 80.67) in comparison to control (47.67) (F = 27.26, df = 2,6, $p < 0.001$) (Fig 2B).

At 60 DAI, analysis of variance (ANOVA) showed a significant increase in fresh and dry shoot and root biomass in seeds treated with Cjc-03 and TS-01 as compared to the control seedlings. Fresh shoots and roots showed significantly higher values for weight (14.14g; 14.21 and 16.74g; 16.81g) of tomato seedlings colonized by Cjc-03 and TS-01 respectively, in comparison to the control plant (7.31g; 7.51g) (shoot of seedling F = 9.67, df = 2,6, $p = 0.013$; seedlings roots F = 4.83, df = 2,6, $p = 0.046$) (Fig 3A and 3B).

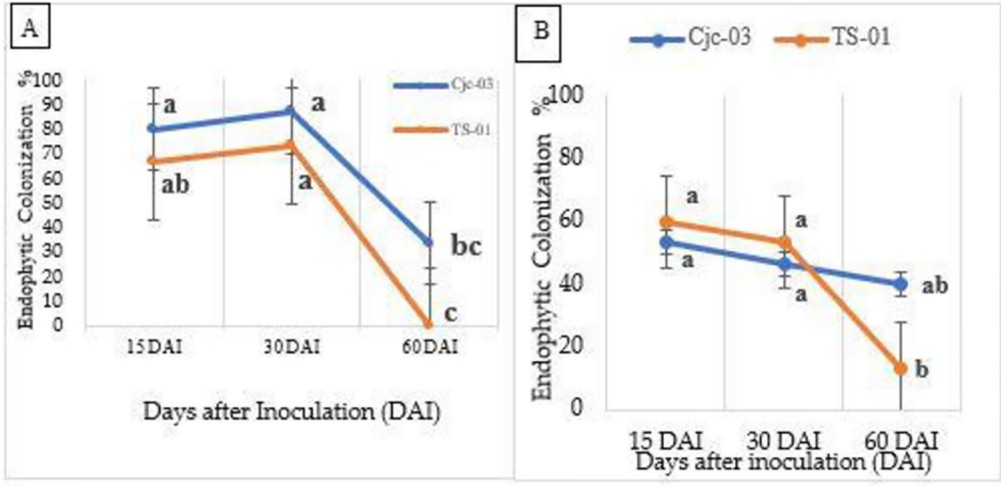

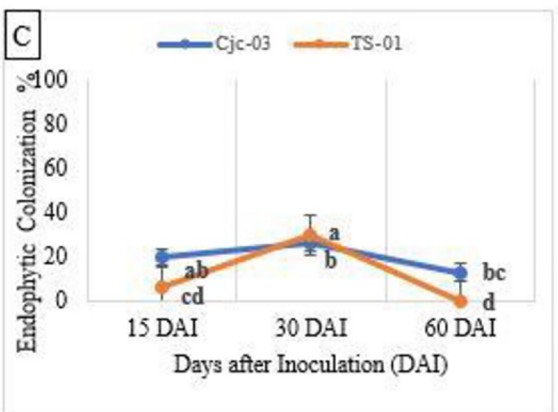

**Fig 1. Endophyte recovery in tomato inoculated with fungal isolates (Cjc-03 and TS-01).** Percentage colonization of different plant parts (A) leaves, (B) stems, and (C) roots of tomato plants inoculated with fungal isolates (Cjc-03 and TS-01) and evaluated at different time after inoculation (15, 30- and 60-days).

The mean value of the dry weight of shoots and roots was significantly higher in fungal colonized plants than in the control plants. (Shoot dry weight: F = 9.1143, df = 2,6, $p$ = 0.015; root dry weight: F = 3.91df = 3,6 $p$ = 0.05) (Fig 4A and 4B).

## Effects of endophytically-colonized tomato seedlings on the mortality, and adult emergence of *B. tabaci*

The endophytic fungal isolates, *I. javanica* (Cjc-03) and *P. lilacinum* (TS-01) significantly increased mortality rates of *B. tabaci* which had fed on colonized tomato seedlings at 7 days after exposure to the aerial parts when compared with those of the control treatment. However, seedlings treated with Cjc-03 caused the highest reduction (51.92±4.78%) of *B. tabaci* than those treated with TS-01 (45.32±0.20%) and there are no significant differences found among the plants treated with two isolates, but these result were significantly differed from those of untreated control seedlings (19.29±2.35), (F = 31.36, df = 2,9, $p$ = < 0.01) (Fig 5A).

Endophytic fungi caused a reduction in adult *B. tabaci* emergence in the first generation. Both isolates caused a significant reduction in the number of adults emerging (35 days post-

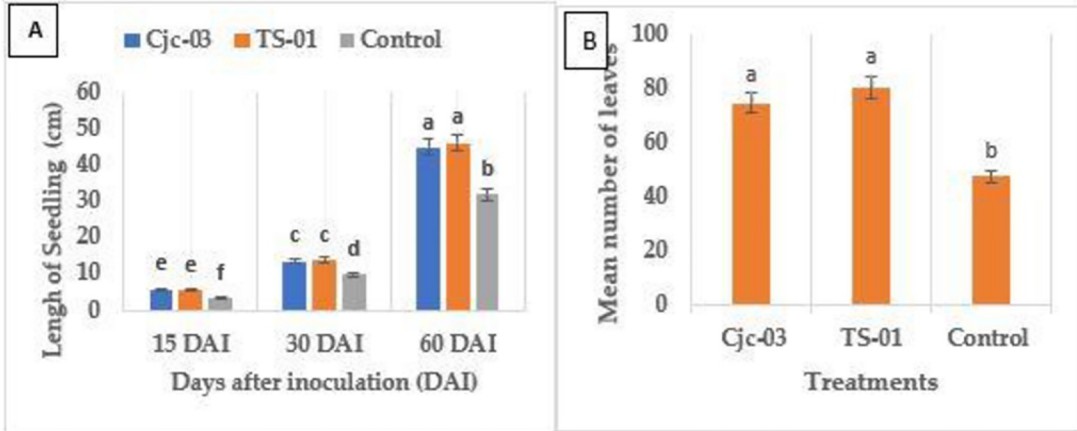

**Fig 2. Mean height (±SE) and number of leaves (leaflets) of tomato seedlings.** (A): Length of tomato plants (shoot) measured at 15, 30, and 60 days after inoculations of tomato seeds with fungal suspension or control. (B): the number of leaves (leaflet) counted at 60 DAI of tomato seeds in fungal suspension or control. The different letters over the bars represent significant differences among the treatments according to Fisher's LSD test (p< 0.05).

exposure). A significantly higher mean number of adults emerge in the control treatment (57.50±2.66) in comparison to the plants treated with Cjc-03 (15.00±1.47) and TS-0 (28.75 ±4.78) (F = 43.89, df = 2,9, *p* = < 0.01) (Fig 5B).

## Discussion

The result of this research indicated that the two EPF isolates tested here have the capacity to colonize tomato plants endophytically which persisted in roots, stems, and leaves until the end of the trials (60 days). In addition, tomato seeds inoculated with *I. javanica*, and *P. lilacinum* isolates significantly promoted plant growth and reducedsurvival of *B. tabaci* adults which had fed on colonized tomato plants and also reduced adult emergence in comparison to control plants.

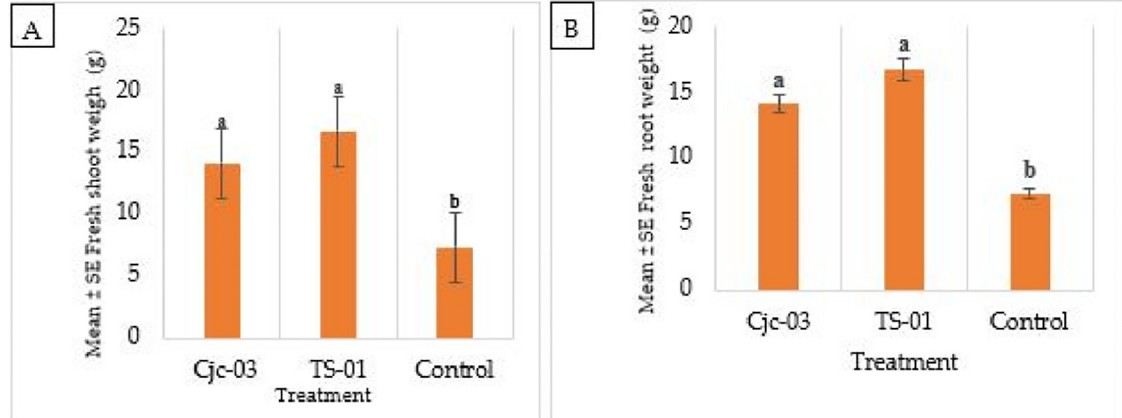

**Fig 3. Mean (± SE) fresh weight (g) of tomato plants inoculated with 2 EPF isolates at 60 DAI.** (A) shoot weight (B), root weight. The different letters over the bars represent significant differences among the treatments according to Fisher's LSD test (p < 0.05).

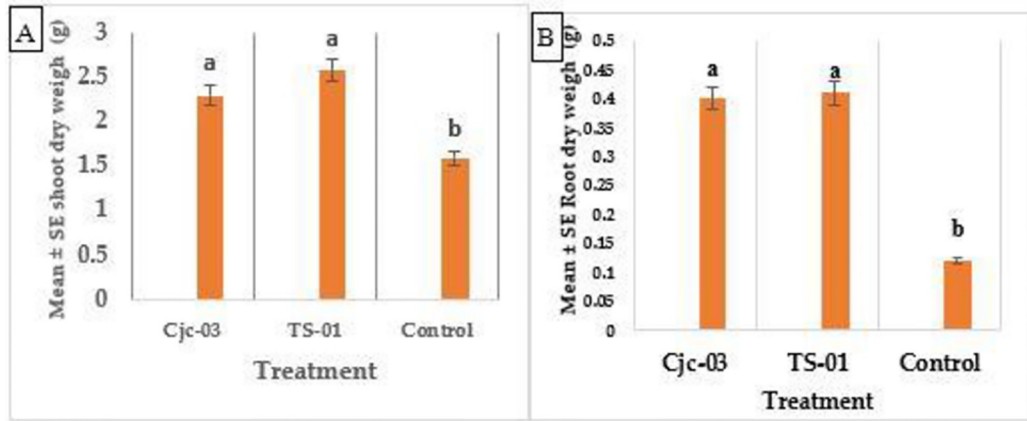

**Fig 4. Mean (± SE) dry weight (g) of tomato plants inoculated with 2 EPF isolates at 60 DAI.** (A) Shoots weight (B), root weight. The different letters over the bars represent significant differences among the treatments according to Fisher's LSD test (p < 0.05).

Previous research findings have reported the successful establishment of *I. javanica*, and *P. lilacinum* as endophytes for the control of insect pests in tomato plants and other economically important plants. Seed treatment by *Isaria* spp. has been successfully shown to be effective for endophytic colonization of pepper [25], eggplant [26], sorghum [27], and English oak [13]. Likewise, inoculation of *P. lilacinum* via seed treatment has been shown to colonize cotton [12, 28].

The percentage of colonization of each isolate was shown to be proportional to the day of assessment with the second period (30 DAI) observed to have the highest percent colonization and the third period (60 DAI) to have the lowest percentage. A similar trend was observed by Mantzoukas et al. [27] who recorded high percent colonization of sorghum leaves and stems

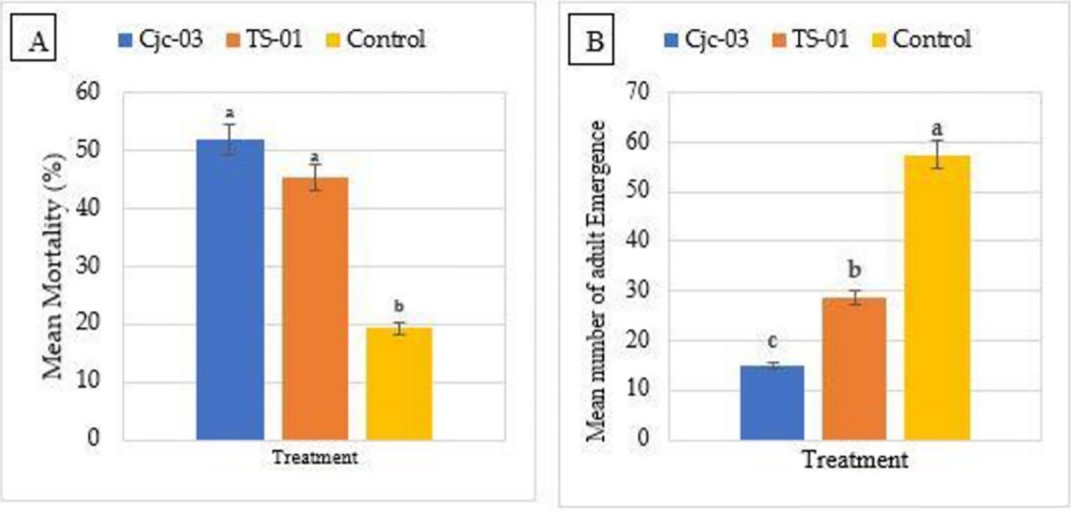

**Fig 5.** Effect of endophytic EPF on the mortality and adult emergence of *B. tabaci* (A): Mean percent mortality (SE ±) of adult *B. tabaci* exposed to tomato seedlings at 21 DAI. (B): Mean number (SE ±) of adult *B. tabaci* emergence on inoculated plants at 35 days after exposing adult *B. tabaci* on inoculated tomato seedlings. Treatments with the same letter were not significantly different according to Fisher's LSD test at *p* = 0.05.

by *B. bassiana*, *M. robertsii*, and *Isaria fumosorosea* at 30 DAI. Similarly, Sun et al. [26] recorded the lowest percentage colonization of plant tissues of eggplant (leaf, stem, and root) by *C. fumosorosea* at 60 DAI.

The current study found that tomato seed inoculated with an isolate of *I. javanica* or *P. lilacinum* promoted growth and increased plant biomass when compared with the uninoculated control plants. There are few reports of plant inoculations with EPF species of the genera *Isaria* and *Purpureocillium* promoting plant growth. For example, Sun et al. [26] reported that colonization of eggplant seedlings by *C. fumosorosea* increased plant height and other growth parameters including shoot length, root length, number of leaves, and weight of fresh shoot and root. However, *P. lilacinum* has been reported to increase the number of nodes and dry biomass of cotton plants [12]. Several factors have shown to enhance plant growth (shoot length, root length number of leaves, and weight of fresh shoot and root) some of them include, the ability to nurture nutrient uptake and to withstand different biotic and abiotic stress [26, 29].

The endophytic colonization of tomato by *I. javanica* and *P. lilacinum* caused significant mortality of whitefly and reduced adult emerge (35 days post-exposure) in comparison with the control plants. Although several studies demonstrated a significant reduction in the number of insect pests on different host plants when inoculated with EPF [4, 7, 8, 30], plant inoculations with fungal pathogens of the genera of *Isaria* and *Purpureocellium* have been rarely reported to cause negative effects on *B. tabaci* and other insect pests. For instance, Sun et al. [26] reported that the inoculation of *C. fumosorosea* isolates in eggplant by seed immersion resulted in a reduction of the *B. tabaci* population. The inoculations of *B. bassiana* in tomato plants by root drenching significantly reduced the number of eggs and nymphs of the greenhouse whitefly, *Trialeurodes vaporariorum* (Westwood) (Hemiptera: Aleyrodidae) in comparison to the uninoculated control plants [8]. Similarly, Lopez and Sword [12], reported a reduction in the population of *Helicoverpa zea* due to the endophytic action of *B. bassiana* and *P. lilacinum* in cotton (*Gossypium hirsutum*).

Dead *B. tabaci* did not show any symptoms of mycosis. It has been previously reported that dead insects found in endophytically colonized plants show no symptoms of fungal infection [10].

## Conclusion

In conclusion, this research provides the first report on the endophytic colonization of plants with EPF and their direct impact on promoting plant growth. Endophytic colonization was detected following seed immersion techniques after inoculating $1 \times 10^8$ fungal suspensions of *I. javanica* (Cjc-03) or *P. lilacinum* (TS-01). Both EPF endophytically colonized tomato seedlings and were recovered during 15, 30, and 60 DAI in the leaves, stems, and roots. Moreover, the plants treated with endophytes had increased height and biomass in comparison to non-treated seedlings. In addition, both endophytic isolates killed whiteflies and reduced *B. tabaci* adult emergence.

## Supporting information

**S1 Fig. Bioassay cages used to evaluate the effect of endophytic EPF against *B. tabaci*.** (A) Cage used to examine the effect of endophytic EPF on *B. tabaci* Population (B) Cage used to examine Adult Emergence of *B. tabaci*.
(DOCX)

**S2 Fig. Examples of endophytic EPF (Cjc-03 and TS-01) re-isolation from the tomato plant tissues.** (A) Leaf, (B) stem, and (C) root tissues following seed treatment with a fungal conidia suspensions.
(DOCX)

**S3 Fig. Representative laser scanning confocal microscopy images of 30 days old tomato plant tissues colonized by endophytic EPF.** (A) Cjc-03 (*I. javanica*) (B) TS-01 (*P. lilacinum*) with the noticeable spores indicated that the spores are alive on the plant tissues; Germination of conidia and appressorium formation in the stem tissues of (C) Cjc-03 (*I. javanica*), (D) TS-01 (*P. lilacinum*).
(DOCX)

**S1 Table. Fungal endophytes used in the study.** Species, accession number, origin, and EPF host species.
(DOCX)

## Acknowledgments

We thank Dr. Razak Terhem and Arisa Azim for technical assistance.

## Author Contributions

**Conceptualization:** Ibrahim Sani, Syari Jamian, Johari Jalinas.

**Data curation:** Norsazilawati Saad, Sumaiyah Abdullah.

**Formal analysis:** Johari Jalinas.

**Funding acquisition:** Syari Jamian, Norsazilawati Saad.

**Investigation:** Ibrahim Sani, Erneeza Mohd Hata.

**Validation:** Syari Jamian, Siti Izera Ismail.

**Writing – original draft:** Ibrahim Sani, Johari Jalinas.

**Writing – review & editing:** Syari Jamian, Siti Izera Ismail.

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
