## [Decision Letter · Decision Letter 0]

5 Dec 2022

PONE-D-22-19772Inoculation and colonization of entomopathogenic fungi, Cordyceps javanica and Purpureocillium lilacinum  in tomato plants, and their effect on seedlings growth, population, and adult emergence of whitefly, B. tabaci (Gennadius).PLOS ONE

Dear Dr. Jamian,

Thank you for submitting your manuscript to PLOS ONE. After careful consideration, we feel that it has merit but does not fully meet PLOS ONE’s publication criteria as it currently stands. Therefore, we invite you to submit a revised version of the manuscript that addresses the points raised during the review process.

We look forward to receiving your revised manuscript.

Kind regards,

Allah Bakhsh

Academic Editor

PLOS ONE

https://journals.plos.org/plosone/s/fileid=ba62/PLOSOne_formatting_sample_title_authors_affiliations.pdf.

“The authors would like to express their gratitude to the Tertiary Education Trust Fund (TETFund), Nigeria for providing a Ph.D. scholarship and Universiti Putra Malaysia (UPM) for the general support during the study.”

 “This study was funded by the Fundamental Research Grant Scheme (FRGS) from the Ministry of Higher Education Malaysia (Grant no. 5540212; reference code FRGS/1/2019/WAB01/UPM/02/36). he funders had no role in study design, data collection and analysis, decision to publish, or preparation of the manuscript.”

3. PLOS requires an ORCID iD for the corresponding author in Editorial Manager on papers submitted after December 6th, 2016. Please ensure that you have an ORCID iD and that it is validated in Editorial Manager. To do this, go to ‘Update my Information’ (in the upper left-hand corner of the main menu), and click on the Fetch/Validate link next to the ORCID field. This will take you to the ORCID site and allow you to create a new iD or authenticate a pre-existing iD in Editorial Manager. Please see the following video for instructions on linking an ORCID iD to your Editorial Manager account: https://www.youtube.com/watch?v=_xcclfuvtxQ.

Reviewers' comments:

Reviewer's Responses to Questions

**Comments to the Author**

1. Is the manuscript technically sound, and do the data support the conclusions?

Reviewer #1: Yes

Reviewer #2: Partly

Reviewer #3: Partly

2. Has the statistical analysis been performed appropriately and rigorously? 

Reviewer #1: Yes

Reviewer #2: Yes

Reviewer #3: Yes

3. Have the authors made all data underlying the findings in their manuscript fully available?

Reviewer #1: Yes

Reviewer #2: Yes

Reviewer #3: Yes

4. Is the manuscript presented in an intelligible fashion and written in standard English?

Reviewer #1: No

Reviewer #2: No

Reviewer #3: Yes

5. Review Comments to the Author

Reviewer #1: The paper is not acceptable in present form. Title: Inoculation and Colonization of Entomopathogenic Fungi, 2 Cordyceps javanica and Purpureocillium lilacinum in 3 Tomato Plants, and their effect on Seedlings Growth, 4 Population, and adult Emergence of Whitefly, Bemisia 5 tabaci (Gennadius)

The authors should give full names of Cordyceps javanica e.g. should be written as when it appears fort he first time and in title e.g. Cordyceps javanica (wf GA17)

Purpureocillium lilacinum shold be written as Purpureocillium lilacinum (Thom) Luangsa-ard, Hou- braken, Hywel-Jones & Samson (2011) when it appears first time and in the title.

There is no problem about control treatments used in the experiment.

Thereafter, the two names could be written as C. javanica and P. lilacinum in the annotated text of the paper. There is need to give more information. The topic is very interesting. Therefore present abstract should be rewritten to give more details about the experiment.

The authors should give more information about the induced conclusions based on the experimental results.

Purpureocillium lilacinum should be written as

Keywords: All words shown in the title must not be shown in the keywords.

İs very long. It shooould be shortened

The authors should not use the language of first or second person. All paper should be written in the language of third person.

Reviewer #2: General comments

This is an interesting study of the endophytic colonization of tomato plants with two of the less studied species of entomopathogenic fungi. Fungal inoculation was performed by treating the seeds with conidia suspensions. The subsequently inoculated seeds were allowed to germinate and various aspects of the plants were then studies: colonization of different parts of the plants from day 15 to day 60 post inoculation; plant development: seedling height/length, number of leaves, fresh and dry weight of roots, shoots and leaves. Furthermore, the authors also looked at some aspects of the interaction of EFP colonized plants with an important tomato pest, B. tabaci. In that case the authors looked at mortality of B. tabaci adults placed on the plants and an adult emergence test, which I could not understand at all.

The results showed the highly positive effects of both fungal isolates on plant development (which has been shown many times before for other EPF endophytes), however, the increased leaf number data is really odd.

The colonization of different parts of the plants was variable, with the highest % in leaves, followed by stems and a low level of colonization in roots. The peak colonization was seen at 30 days, falling rapidly at 60 days for leaves. The pattern for stem colonization was different to that of leaves, with the highest levels at 15 days, dropping off at 30 days.

Importantly, the authors used confocal microscopy to look at plant colonization. However, they give little attention to these results and put the images in the supporting material. In fact, the data for microscopy needs explaining as it is rather confusing, but maybe it is worth exploring in more detail as very few endophyte studies have used this technique.

The data for insect-plant interactions is also very confusing, with very little explanation of the M&M or results.

I have made some corrections to the PDF file, but the English needs a general overhaul.

Specific comments

Modify the title as very confusing!

See attached file.

Reviewer #3: The hypothesis of the work is wonderful. Eventhough, some monor corrections/typos needs to be corrected. However, it is not clear that how the auther has conducted the mortality experiments? How the experiment was conducted ? the the adults were relased and observed? simultaneously how the emergence observation were recorded. This methodology part need to be described well so as to further understand the esults and conclusion.

6. PLOS authors have the option to publish the peer review history of their article (what does this mean?). If published, this will include your full peer review and any attached files.

Reviewer #1: **Yes: **The paper should be written in the language of third person. moreover following information is also needed

Title: Inoculation and Colonization of Entomopathogenic Fungi, 2 Cordyceps javanica and Purpureocillium lilacinum in 3 Tomato Plants, and their effect on Seedlings Growth, 4 Population, and adult Emergence of Whitefly, Bemisia 5 tabaci (Gennadius)

The authors should give full names of Cordyceps javanica e.g. should be written as when it appears fort he first time and in title e.g. Cordyceps javanica (wf GA17)

Purpureocillium lilacinum shold be written as Purpureocillium lilacinum (Thom) Luangsa-ard, Hou- braken, Hywel-Jones & Samson (2011) when it appears first time and in the title.

There is no problem about control treatments used in the experiment.

Thereafter, the two names could be written as C. javanica and P. lilacinum in the annotated text of the paper. There is need to give more information in abstract about the experiment. The topic is very interesting. Therefore present abstract should be rewritten to give more details about the experiment.

The authors should give more information about the induced conclusions based on the experimental results.

Purpureocillium lilacinum should be written as

Keywords: All words shown in the title must not be shown in the keywords to avoid duplication of key words. the title is very long. It should be shortened.

The authors should not use the language of first or second person. All paper should be written in the language of third person.

Reviewer #2: No

Reviewer #3: **Yes: **Satish Kumar Sain

---

## [Author Response · Author response to Decision Letter 0]

16 Jan 2023

The authors are very much thankful to the reviewers for their meticulous review and valuable comments. We have now incorporated all the two reviewers’ comments and suggestions to improve the manuscript. Details of our response to each comment, corrections and suggestions are given in a separate file title "response of reviewer comments". However, funding information have been removed in the acknowledgments section and has been corrected as directed.

---

## [Decision Letter · Decision Letter 1]

10 Feb 2023

PONE-D-22-19772R1Inoculation and colonization of the entomopathogenic fungi, Isaria javanica and Purpureocillium lilacinum, in tomato plants, and their effect on seedling growth, mortality and adult emergence of  Bemisia tabaci (Gennadius)PLOS ONE

Dear Dr. Jamian,

Thank you for submitting your manuscript to PLOS ONE. After careful consideration, we feel that it has merit but does not fully meet PLOS ONE’s publication criteria as it currently stands. Therefore, we invite you to submit a revised version of the manuscript that addresses the points raised during the review process.

We look forward to receiving your revised manuscript.

Kind regards,

Allah Bakhsh

Academic Editor

PLOS ONE

Reviewers' comments:

Reviewer's Responses to Questions

**Comments to the Author**

1. If the authors have adequately addressed your comments raised in a previous round of review and you feel that this manuscript is now acceptable for publication, you may indicate that here to bypass the “Comments to the Author” section, enter your conflict of interest statement in the “Confidential to Editor” section, and submit your "Accept" recommendation.

Reviewer #1: All comments have been addressed

Reviewer #2: (No Response)

Reviewer #3: All comments have been addressed

2. Is the manuscript technically sound, and do the data support the conclusions?

Reviewer #1: Yes

Reviewer #2: Yes

Reviewer #3: Partly

3. Has the statistical analysis been performed appropriately and rigorously? 

Reviewer #1: Yes

Reviewer #2: No

Reviewer #3: N/A

4. Have the authors made all data underlying the findings in their manuscript fully available?

Reviewer #1: Yes

Reviewer #2: Yes

Reviewer #3: Yes

5. Is the manuscript presented in an intelligible fashion and written in standard English?

Reviewer #1: Yes

Reviewer #2: No

Reviewer #3: Yes

6. Review Comments to the Author

Reviewer #1: I have carefully read and evaluated the paper. I found and recommend that the paper should be accepted and published.

Reviewer #2: It is somewhat tiring to re-review a manuscript where the authors haven’t taken into consideration your request to at least improve the English language beyond the corrections I marked in the manuscript. I have made some more corrections but many phrases are poorly constructed and the authors obviously did not ask for any help from a native speaker who understands the area. The description of the adult emergence experiment is still poor and the terminology for the adult survival experiment is also incorrect (population reduction: would be in the field but in a greenhouse, it is mortality or survival).

Reviewer #3: The name of the EPF is still Isaria javanica in place of Cordyceps javanica which is the latest name. I do not understand why this big mistake has happened in the revised MS

7. PLOS authors have the option to publish the peer review history of their article (what does this mean?). If published, this will include your full peer review and any attached files.

Reviewer #1: **Yes: **the review is done and the paper is evaluated by myself

Reviewer #2: No

Reviewer #3: **Yes: **Satish Kumar Sain

---

## [Author Response · Author response to Decision Letter 1]

2 Mar 2023

The authors are very much thankful to the reviewers for their meticulous review and valuable comments. We have now incorporated all the two reviewers’ comments and suggestions to improve the manuscript.

---

## [Decision Letter · Decision Letter 2]

20 Apr 2023

PONE-D-22-19772R2Inoculation and colonization of the entomopathogenic fungi, Isaria javanica and Purpureocillium lilacinum, in tomato plants, and their effect on seedling growth, mortality and adult emergence of  Bemisia tabaci (Gennadius)PLOS ONE

Dear Dr. Jamian,

Thank you for submitting your manuscript to PLOS ONE. After careful consideration, we feel that it has merit but does not fully meet PLOS ONE’s publication criteria as it currently stands. Therefore, we invite you to submit a revised version of the manuscript that addresses the points raised during the review process.

We look forward to receiving your revised manuscript.

Kind regards,

Allah Bakhsh

Academic Editor

PLOS ONE

Journal Requirements:

Reviewers' comments:

Reviewer's Responses to Questions

**Comments to the Author**

1. If the authors have adequately addressed your comments raised in a previous round of review and you feel that this manuscript is now acceptable for publication, you may indicate that here to bypass the “Comments to the Author” section, enter your conflict of interest statement in the “Confidential to Editor” section, and submit your "Accept" recommendation.

Reviewer #2: (No Response)

2. Is the manuscript technically sound, and do the data support the conclusions?

Reviewer #2: Yes

3. Has the statistical analysis been performed appropriately and rigorously? 

Reviewer #2: Yes

4. Have the authors made all data underlying the findings in their manuscript fully available?

Reviewer #2: Yes

5. Is the manuscript presented in an intelligible fashion and written in standard English?

Reviewer #2: Yes

6. Review Comments to the Author

Reviewer #2: I have corrected your manuscript yet again. I guess it is ok now but please stop using the term population for a group of insects in a cage or pot.

7. PLOS authors have the option to publish the peer review history of their article (what does this mean?). If published, this will include your full peer review and any attached files.

Reviewer #2: No

---

## [Author Response · Author response to Decision Letter 2]

25 Apr 2023

The authors are very much thankful to the reviewers for their meticulous review and valuable comments. We have now incorporated all the

reviewers’ comments and suggestions to improve the manuscript.

---

## [Editor Report · Decision Letter 3]

28 Apr 2023

Inoculation and colonization of the entomopathogenic fungi, Isaria javanica and Purpureocillium lilacinum, in tomato plants, and their effect on seedling growth, mortality and adult emergence of  Bemisia tabaci (Gennadius)

PONE-D-22-19772R3

Dear Dr. Jamian,

We’re pleased to inform you that your manuscript has been judged scientifically suitable for publication and will be formally accepted for publication once it meets all outstanding technical requirements.

Kind regards,

Allah Bakhsh

Academic Editor

PLOS ONE
---

## [Editor Report · Acceptance letter]

9 May 2023

PONE-D-22-19772R3 

Inoculation and colonization of the entomopathogenic fungi, *Isaria javanica
* and *Purpureocillium lilacinum,* in tomato plants, and their effect on seedling growth, mortality and adult emergence of Bemisia *tabaci* (Gennadius) 

Dear Dr. Jamian:

I'm pleased to inform you that your manuscript has been deemed suitable for publication in PLOS ONE. Congratulations! Your manuscript is now with our production department. 

Kind regards, 

on behalf of

Dr. Allah Bakhsh 

Academic Editor

PLOS ONE